# GPS-Based Indoor/Outdoor Detection Scheme Using Machine Learning Techniques

**Van Bui, Nam Tuan Le, Thanh Luan Vu, Van Hoa Nguyen and Yeong Min Jang ***

Department of Electronics Engineering, Kookmin University, Seoul 02707, Korea; buivandut@gmail.com (V.B.); namtuanlnt@gmail.com (N.T.L.); vuthanhluan999@gmail.com (T.L.V.); vanhoahd95@gmail.com (V.H.N.)

**\*** Correspondence: yjang@kookmin.ac.kr; Tel.: +82-2-910-5068



**Featured Application: Supporting commercial mobile applications, improving indoor channel estimation and mobile performance in indoor environment. Applied in Optical Camera Communication (OCC) applications to detect the outdoor environment.**

**Abstract:** Recent advances in mobile communication require that indoor/outdoor environment information be available for both individual applications and wireless signal transmission in order to improve interference control and serve upper-layer applications. In this paper, we present a scheme to identify the indoor/outdoor environment using GPS signals combined with machine learning classification techniques. Compared to traditional schemes, which are based on received signal strength indicator (RSSI), the proposed scheme promises a robust approach with high accuracy, smooth operation when moving between indoor and outdoor environments, as well as easy implementation and training. The proposed scheme combined information from a certain number of GPS satellites, using the GPS sensor on mobile devices. Then, data are collected, preprocessed, and classified as indoor or outdoor environment using a machine learning model that is optimized for the best performance. The GPS input data were collected in the Kookmin University area and included 850 training samples and 170 test samples. The overall accuracy reached 97%.

**Keywords:** I/O detection; GPS signal; machine learning; positioning applications.

## 1. Introduction

Recent developments in Mobile and mobile communication means that an excessive amount of information about the working environment is required for both individual applications and wireless signal transmission. One type of information is indoor/outdoor detection (I/O detection). I/O detection refers to the estimation of the mobile user's working condition, i.e., whether the device is being operated in an outdoor or indoor environment. This information is beneficial in many mobile applications and wireless communication. However, few studies focus on I/O detection.

I/O detection can provide useful information about user behavior and the appropriate use of mobile network resources [1,2]. I/O detection can also provide essential information for the upper layer of mobile applications. For example, a Global Positioning System (GPS) location management system can use this information to turn a GPS off to save power. GPSs only work well in outdoor environments; therefore, turning the system off GPS in indoor environments can save energy for other applications [3]. Another example where I/O detection can improve energy consumption is Wi-Fi access point searching behavior. Typically, indoor environments provide more access points as well as stronger signals. I/O detection can reduce energy consumption of both GPSs and Wi-Fi connection applications, as well as mobile behaviors. I/O detection can help a mobile device adapt to an indoor environment. For example, I/O detection can facilitate adjusting the device volume, trigger

reminders, changing the device's home screen, and adjusting screen brightness. I/O detection also plays an important role in indoor channel estimation. Indoor channels block some radio signal and have significant interference noise. I/O detection can improve channel estimation prediction as well as data transmission [4]. I/O detection can be used in Optical Camera Communication (OCC) applications since the outdoor environment decreases the OCC performance. Based on the I/O information, OCC applications will have more information to adapt and improve its performance.

Some GPS constellations consist of 31 satellites, of which 27 are used [5]. GPS satellite systems, produced and maintained by the United States of America, provide an efficient way to detect a mobile user's location, sometimes within centimeters. GPS signals are always available and provide a reliable working environment and low-energy consumption compared to other communication systems, such as Wi-Fi or 3G. However, GPS signals from a single satellite are unstable, vary from location to location, and can change when the user moves or when the signal is occluded. The number of available satellites from which a GPS sensor receives signals changes over time and cannot be determined without a complex calculation. In this paper, GPS signals will be synthesized to provide information for I/O detection applications. The proposed I/O detection method for mobile devices has low-power consumption, easy implementation, and flexible operation compared to other I/O detection methods.

As mentioned previously, raw GPS signals contain information about indoor/outdoor condition; however, indoor/outdoor condition information is difficult to synthesize and calculate. Currently, machine learning, which is attracting a great deal of attention as a popular classifier technique, provides a robust solution for I/O detection problems. The advantages of machine learning compared to other classification methods are its high accuracy and ability to handle all types of input data. The redundant GPS signal data is the best condition for machine learning to perform effectively, despite the complex structure of GPS input data. In this paper, we provide a supervised algorithm as the primary solution because it is more suitable for classifier tasks. The supervised machine learning family includes support vector machines (SVM), logistic regression technique, naive Bayes classifiers, k-nearest neighbor (KNN) technique, and neural networks. A machine learning approach provides a simple solution, low-power costs, and high accuracy for I/O detection problems.

The proposed scheme based on the GPS signal and machine learning methods require a very small volume of dataset, as only 850 training samples are enough for an accuracy classification. Furthermore, a mobile phone application can apply offline learning, which uses the trained model for the classification and updated if requested, so it does not require a training dataset. Therefore, the proposed scheme will be suitable for almost all mobile applications that require I/O environment information.

The remainder of this paper is organized as follows. Section 2 describes related work. We discuss GPS signal characteristics and how we can take advantage of them in Section 3. Section 4 provides the working scheme for an I/O detection scheme, including the data collection process, preprocessing, and the training and testing phases. The supervised evaluation results will be discussed in Section 5. The paper will be concluded in Section 6.

## 2. Related Work

Current research into I/O detection is very limited and many studies in this field are primarily based on image processing, which is not suitable for mobile applications due to the complex calculation capacity and large memory required for input data [6–8]. However, there are practical methods to deal with this problem. One such method is based on collecting data using various sensors that are available on mobile devices [1,9–12]. Another is based solely on collecting data using a GPS sensor [13,14].

In a previous study [9], the authors proposed an I/O detection method that used a set of mobile sensors, including a light sensor, a cellular module, and a magnetic sensor for I/O detection. That proposed method promised high accuracy and energy efficiency. This combination of sensors provided high accuracy (88%–90%) for both indoor and outdoor environments with average power consumption. Although the input data is available with mobile, this method requires complex calculations and a large amount of input data to work well. In another study, the author used even more sensors and data for

I/O detection, including an activity recognition Application Programming Interface (API), barometric pressure, a light sensor, cloud coverage data, a timer, Global System for Mobile Communications (GSM) signal strength, an accelerometer, a magnetometer, and ambient noise [10]. These sets of input data increase I/O detection accuracy to more than 98%; however, calculation and data collection are complicated and consume significant power.

Harsh testing conditions also affect I/O detection accuracy. In [11], the authors used the same input dataset with previous study [10], including an accelerometer, cellular radio, light sensor, magnetometer, and proximity sensor data; however, the accuracy is only 80%. Power consumption is also greater compared to a previous study [9]. Primarily based on light sensor data and a Wi-Fi sub-detector with a machine learning approach, another [12] obtained more than 90% accuracy. However, the training model is very complicated and difficult to implement. Similarly, reference [1] uses a machine learning approach for data processing and obtained high accuracy. Again, data consumption and a complicated calculation are disadvantages of this approach.

Another approach is using only GPS signal because it is always available and provides significant information for I/O detection. In addition, this approach consumes less power compared to Wi-Fi and 3G signals. In a study about I/O detection based only on sparse GPS positioning information, Iwata et al. obtained 98% accuracy for I/O detection [14]. Although the test sample was very small (53 samples) and the approach required significant time and a complex calculation, this study promised highly accurate I/O detection based on GPS signals and a machine learning classifier. Based on a direct analysis of GPS signals, Chen et al. provided three GPS-based techniques for I/O detection [13]. The technique that provided the best performance was SatProbe, which searches the raw signals from all available GPS satellites to identify input data patterns. With this technique, using a sizable dataset, accuracy reached 86% and it worked much better than the traditional method. Although the accuracy is not too high, as well as the complex calculation, this study also emphasizes the importance of raw GPS signals in I/O detection problems.

A breakthrough approach for the indoor classification and localization is the combination of GPS and Wi-Fi signal, which is a flexible method to deal with emergency situations, where the GPS signal becomes unstable and weaker. In particular emergency situations, when the transmission of the GPS signal was obstructed, the localization process is impossible with GPS signal only, the Wi-Fi combination will be helpful in both indoor/outdoor detection and localization. Reference [15] provides a robust system for tracking GPS in such situation, that the Navigation System can be based on the Wi-Fi signal to improve the accuracy of uses. The same concept can be applied in the I/O detection scheme and improve its performance in the emergency situation, when the GPS signal becomes unstable and weaker.

## 3. Data Analysis

### 3.1. Global Positioning System

The GPS is a global navigation satellite system that geolocation information to GPS receivers [16]. GPS is a satellite-based radio navigation system; consequently, solid obstacles, such as mountain and buildings, weaken and can even block GPS signals. The GPS does not require any transmission data from users, and its operation is independent from telephonic or internet signal. Currently, the GPS is owned by the United States' government and operated by the United States Air Force. The United States' government is responsible for creating and maintaining the system and makes it freely accessible to any GPS receiver [17].

The GPS calculates the receiver's position using the time and position of the satellite system. The satellites use stable atomic clocks that are synchronized all satellites and with the ground receivers. Any offset from GPS time of ground receiver is corrected daily. Similarly, satellite locations are controlled with great accuracy. Each GPS satellite continuously transmits a radio signal to provide the current time and its position; therefore, GPS sensors can receive signals without issuing a request. The distance

between a satellite and a GPS receiver is calculated based on the delay time of the transmitted and received signals due to the constant speed of radio waves. A GPS receiver updates the distance of multiple satellites and determines the accurate position of the receiver and its deviation from real time. At least four satellites must be used to compute four unknown quantities and solve the accuracy equation [5].

The GPS satellite system comprises 24–32 satellites in medium Earth orbit. Originally, the GPS had only 24 satellites. In that configuration, each set of eight satellites were in three approximately circular orbits [18]. However, the system was modified to six orbital planes with only four satellites each (Figure 1). The orbital period is one-half a day (11 h and 58 min) so that the satellites pass over the same locations every day. The satellite orbits are carefully arranged so that at least six satellites are in line of sight from anywhere on the Earth's surface [5].

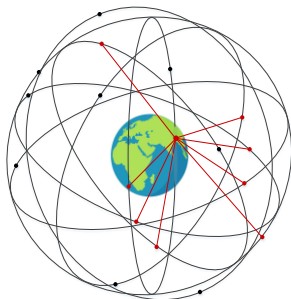

**Figure 1.** Illustration of a 24 GPS satellite constellation in motion relative to the Earth's rotation (Source: Wikipedia [5]).

As of February 2019, there were 32 GPS satellites in orbit, and 27 satellites are in use [19]. Increasing the number of satellites improves GPS sensor calculations by providing redundant measurements, which leads to better position estimation. With the addition of redundancy satellites, the constellation was changed to a non-uniform arrangement. Compared to a uniform arrangement, a non-uniform arrangement dramatically improves the reliability and availability of the GPS system if multiple satellites malfunction [19]. At any given time, at least nine satellites can be observed from a single point on the ground, which is significantly more than the minimum four satellites required for accurate positioning.

Other satellite navigation systems, including GLONASS (GLObal Navigation Satellite System) developed by Russia and GALILEO developed by Europe [20], can be used in the I/O detection applications as an alternative option besides the GPS. The combination of GPS and other navigation systems can also be a promising approach, since it increases the reliability in both navigation [21] and I/O detection. However, the requirement of I/O detection is different from navigation systems since it prefers the unification of satellites over the number of it. Using only GPS signal guarantees a united system, signal, and the number of satellites, that can be adopted in all mobile phones nowadays. The other reason is that I/O detection does not require as much information as the navigation system, and the redundancy of GPS satellites is enough for the application.

In this study, we use two characteristics of raw GPS signals for I/O detection. The first is GPS signal redundancy. Assuming no occluding obstacles, approximately nine satellites can be observed from any one point on the ground. Consequently, GPS signals can always be received with desirable signal strength. The second characteristic is the fact that the GPS satellite navigation system is radio-based. Therefore, solid obstacles, such as mountains and tall buildings, can weaken and even block GPS signals. We used this characteristic to identify if the sensor is in an indoor or outdoor environment.

### 3.2. Data Analysis

As mentioned previously, raw GPS signals vary relative to time and place, which makes the input data complex and difficult for analyzing. The machine learning technique requires simpler input data

in order to extract unique features and find similar patterns. Therefore, in this study, the preprocessing phase is the most important part. We will investigate some features of raw GPS signals to identify the best way to deal with input data.

- GPS signals are always available.

Of the 31 GPS satellites in Earth orbit, 27 are in use. The minimum number of GPS satellites required for normal operation is 24, which means that at least seven satellites are redundant. In practice, in a defined position on the Earth's surface, at least nine satellites always available to send GPS signals to receivers. Due to this redundancy, receivers can obtain stable GPS signals anywhere on Earth, and at least two or three signals will be sufficiently strong for data collection and analysis. However, the satellites always moving; thus, tracking signals from a fixed GPS satellite is impractical. A large number of satellites is also a challenge. The transmission satellites ID changes frequently; therefore, GPS sensors cannot obtain a fixed set of satellite IDs at one time. Due to these problems, a practical approach is to measure all GPS satellites signals and ignore the individual IDs. This technique will provide at least two or three strong GPS signals in an outdoor environment and weak signals in an indoor environment in an unorganized order.

- GPS signal transmission is radio-based.

The GPS is a satellite-based radio navigation system; therefore, solid obstacles, such as mountains and tall buildings, can weaken and even block the signal. If the GPS sensor is in an indoor environment, the signal will be weakened and may even disconnect (Figure 2a). Based on this characteristic, indoor environments will be detected easily because the GPS signal is weakened dramatically. In the indoor environment, the GPS signal is weakening, so the navigation cannot be done with lacking information. However, that is not the case of proposed I/O detection scheme, since it based on the weakened signal of GPS satellites to identify the indoor environment. Detecting signals in an outdoor environment is more difficult because the signal may be blocked by tall buildings or mountains (Figure 2a). As shown in Figure 2b, the GPS signal is blocked by a tall building. Therefore, if we use signals from that satellite, we cannot determine if the device is indoors or outdoors. A similar situation is shown in Figure 3. Here the GPS sensor can receive signals from a satellite, even indoors. Considering these problems, we investigated another approach and determined that stronger GPS signals are more important than weak signals, and the number of GPS signals we collect is not significant.

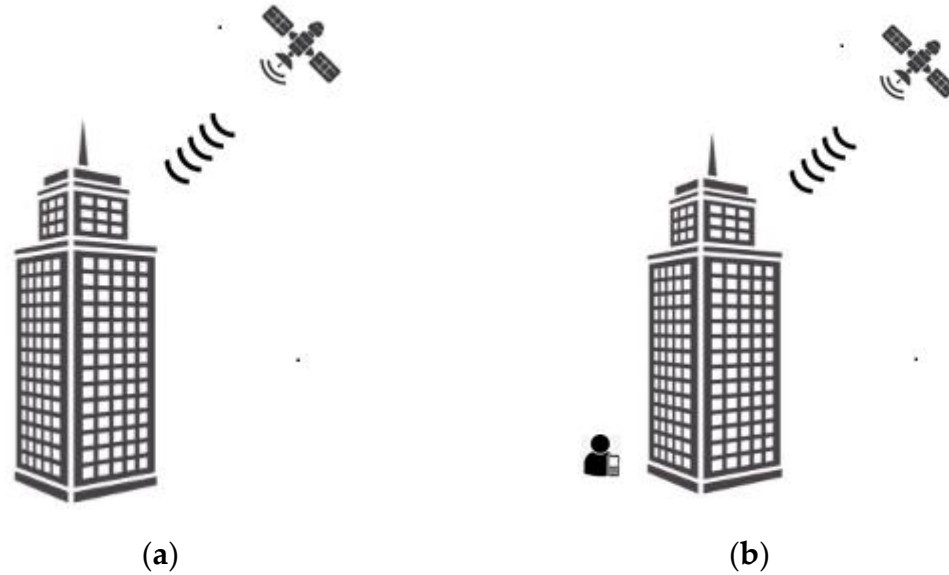

(**a**)  (**b**)

**Figure 2.** (**a**) GPS signal is blocked inside the building and (**b**) GPS signal is blocked by a tall building.

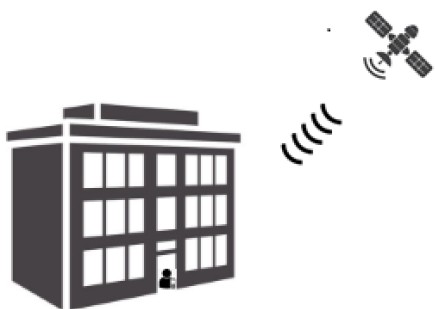

**Figure 3.** GPS sensors can receive signals when the receiver is close to an external door.

- Every mobile device has a GPS sensor.

　　Every mobile device has a GPS sensor that receives many active GPS positions constantly. Therefore, the input data for an I/O detection application is always redundant. In addition, GPS sensors are consistent among mobile devices; thus, we can apply offline learning for our application, leading to a simple and fast calculation model.

- Machine learning techniques are most suitable for I/O detection.

　　As mentioned previously, machine learning is a robust classifier technique to identify the input pattern on I/O detection applications. Machine learning has high accuracy, can use all types of input data, and is adaptable to complex input patterns. Furthermore, compared to traditional approaches, redundant GPS input data also increases the accuracy of machine learning techniques. Another advantage of ML is the diversity of ML techniques, which allows us to select the best technique to address our I/O detection problem. Finally, offline learning for a mobile application is easy to implement and low cost. In addition, calculations are performed quickly, and offline learning does not require a large amount of input data.

## 4. I/O Detection Scheme

　　The proposed scheme is shown in Figure 4: firstly, signal received from GPS satellites was collected through a specialized Mobile application. After collected, the raw input was reprocessed to filter out the errors and missing data. Input data was then analyzed and classified by various machine learning techniques to find the best solution for the I/O detection problem. For special cases of semi-indoor environments, we proposed another preprocessing process to increase the accuracy of proposed scheme. By using different types of machine learning techniques, the proposed scheme becomes flexible to deal with various problems in real life. Furthermore, the proposed preprocessing process will increase the overall accuracy in the semi-indoor environment, which is hard to deal with.

*4.1. Data Collection*

　　We used a mobile application to collect and preprocess the raw GPS signal information. This application collects information from GPS sensors, rearranges the information, and analyses it for better input data. The mobile application was selected for data collection and preprocessing rather than a GPS sensor module because it is simple, suitable for mobile devices, and provides stable input data. In Figure 5, the number and the magnitude of GPS signals in orbit is determined and preprocessed for easier access for data collection. The satellite ID is given below each GPS signal; however, the ID is ignored during the data collection phase. The magnitude of all GPS signals will be recorded and used as input data for the machine learning classifier.

　　In this study, we analyzed statistical differences by focusing on the difference between GPS signals received in indoor and outdoor environments. A large dataset comprising real signal values collected

at multiple locations and in various environments was used. We investigated the impact of indoor, semi-indoor, and outdoor environments on the received GPS signal.

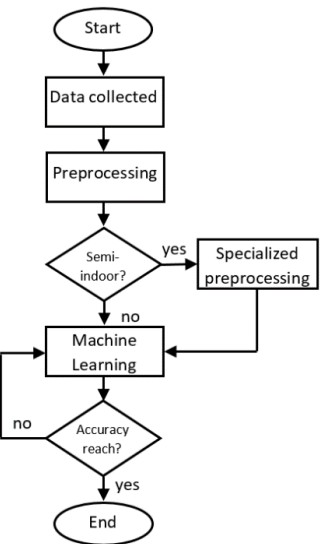

**Figure 4.** Proposed scheme.

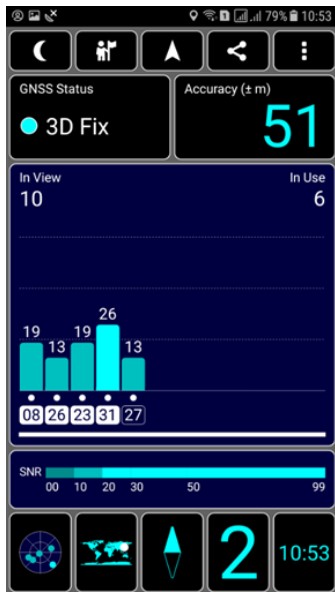

**Figure 5.** GPS signal determination application.

### 4.1.1. Data Collection

The input dataset consists of received GPS signals read directly from the GPS Test application after the preprocessing phase. For convenience, the signals were analyzed and transformed into numbers. Every received GPS signal has an ID number that corresponds to the sending GPS satellite. The magnitude of GPS signals varied from 6 to 99 to indicate signal strength, i.e., stronger signals have greater magnitude. Signals with magnitude less than 6 were not collected and forced to a default value of zero. This approach can reduce noise probability and decrease the number of satellite signals that cannot be used as an anchor. As described previously, we only needed a maximum of nine GPS signals per set. After collecting, each set of GPS signals was sorted from strongest to weakest. This data was checked again and used as input data for I/O detection problems.

The signals were collected in a three-week period from 2 to 23 May 2019. Each sample was obtained in a different location within a 10 km² area. The collection area included many different environments, such as buildings, restaurants, urban areas, hills, loose forests, and basements. The data collection area is a complex urban area with tall building and other obstacles, such as upper highways and mountains. These obstacles block GPS signal from different directions, which makes the dataset varied and difficult to predict. The buildings also have different architectures. Some architectures block GPS signal and other allow the signal to pass into or through the building. The collection period occurred during spring in Korea. Therefore, we were able to collect data under various types of weather conditions, such as foggy, sunny, windy, and rainy. The collection period occurred from 07:00 to 21:00. This time period reflects the changing position of GPS satellites moving in Earth orbit. These conditions were designed to reflect the complexity and variety of a real-world mobile user.

The set of indoor environment signals were collected in the Kookmin University's Campus and the surrounding area (Figures 6 and 7). The largest datasets were collected in the College of Engineering building (number 3), which, compared to other buildings, is large and isolated. The semi-indoor dataset was primarily collected mainly in Building 7 (number 14) and the College of Business Administration (number 12). These two buildings have a large number of windows and some places have glass walls, which GPS signals can pass through. The other dataset was collected in Bugak Hall (number 2), the College of Law (number 4), the College of Design (number 5), a gymnasium (number 6), the International Hall (number 11), the Student Union building (number 15), the Global Center (number 17), and a dormitory (number 21). Based on the various locations, we are confident that the data covers the overall indoor environment characteristics.

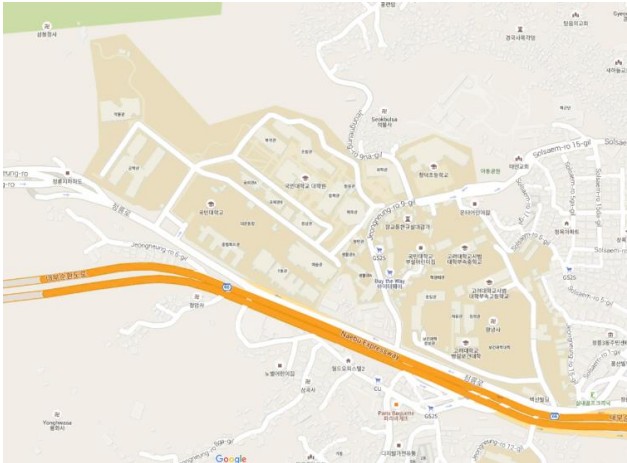

**Figure 6.** Survey area, includes Kookmin University Campus and the surrounding area.

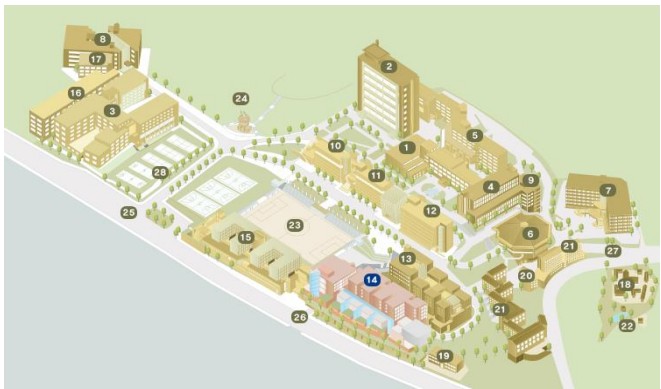

**Figure 7.** Kookmin University's Campus.

The set of outdoor environments were also collected in Kookmin University's Campus, as well as the surrounding urban area and highways. The data collection was large and included various types of environments, some of which had high obstacles that can block and weaken GPS signals (Figure 8).

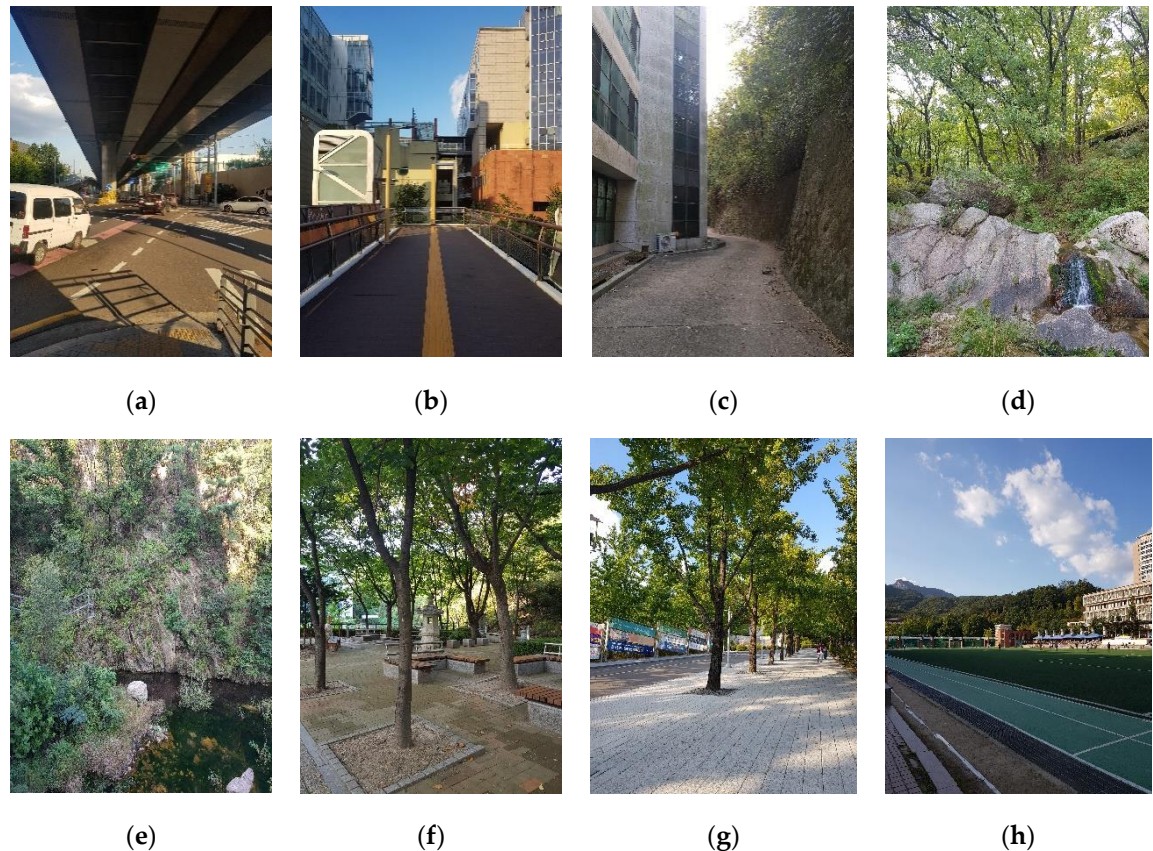

|       |       |       |       |
|:-----:|:-----:|:-----:|:-----:|
| (**a**) | (**b**) | (**c**) | (**d**) |
| (**e**) | (**f**) | (**g**) | (**h**) |

**Figure 8.** Outdoor environment for data collection: (**a**) under a highway, (**b**) walking road, (**c**) high obstacles, (**d**) thin forest, (**e**) beside a mountain, (**f**) University campus, (**g**) street, (**h**) playground.

Another scenario that can be considered is the vehicular application when vehicles access urban canyons that could significantly impair signal reception from navigation satellites. The test set included a high building similar to the urban canyons and also the parking ground. In Figure 9, the GPS signals are different between the underground parking lot and the outside environment. According to this scenario, the I/O detection can be applied to the vehicular application. I/O detection can classify the indoor parking lot and the outside environment. Due to this characteristic, the proposed scheme can improve the quality of the Navigation System. I/O detection scheme can provide information of places where the navigation system required an enhancement method, such as the combination of Wi-Fi signal [15] or other GPS systems [20,21] to improve its accuracy.

### 4.1.2. Data Description

The collected data comprise GPS signals measured by the mobile phone application (GPS signal magnitude mode) and recorded. The dataset described in this subsection was collected using this mode.

Figure 9 also shows the GPS signal magnitude obtained with the dataset. The significant offset between indoor and outdoor values is the result of the substantial difference and attenuation variation in radio signal propagation due to wall or roof obstacles in the indoor environment. However, in some cases, this difference overlaps due to variation in the environment. Each data is denoted as vector X, with components $x_1, x_2 \ldots x_i$, where $x_i$ is a set with nine integer numbers. The dataset includes 1020 vectors X and 1020 corresponding classes Y. Training and test sets were collected without simulation or

adding noise. Note that some data were collected from a semi-indoor environment (this problem will be discussed later).

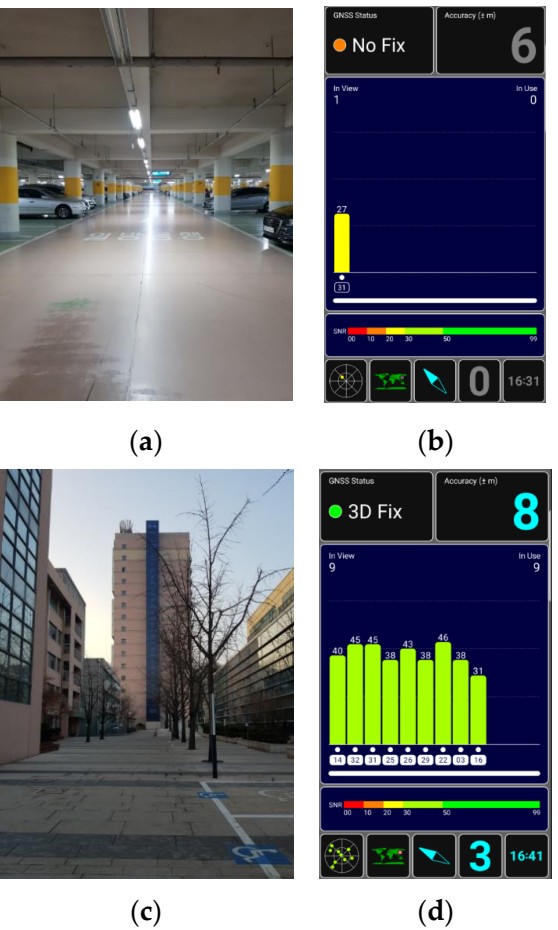

**Figure 9.** Data collected similar to a traffic environment: (**a**) underground parking, (**b**) corresponding GPS signal, (**c**) outdoor environment, (**d**) corresponding GPS signal.

*4.2. Machine Learning Approaches*

4.2.1. Logistic Regression

Logistic regression is a machine learning technique to estimate the parameters of a logistic model, where the logarithm of the equation with a value labeled as "1" is a linear combination of one or more independent variables. The probability of a value being labeled "1" can vary between 0 and 1, and the function that converts the equation to a probability is the logistic function.

Logistic regression is useful in various fields, such as data analysis, medical, economics, and social sciences. The advantages of logistic regression are simplicity, ease of use and calculation, applicability to any type of inputs, and identification of the probability of output values. However, logistic regression is unsuitable for complex problems. In addition, it is sensitive to noise and can easily suffer from overfitting.

Consider a model with two variables $x_1$, $x_2$ and one binary response Y, which have probability $p = P(Y = 1)$. Consider a linear relationship between the variables and the log-odds of $p$ that Y = 1. This linear relationship can be expressed as follows:

$$l = \log_b \frac{p}{1-p} = \beta_0 + \beta_1 x_1 + \beta_2 x_2,\tag{1}$$

where $\ell$ is the log-odds, $b$ is the base of the logarithm, and $\beta_i$ represents the model's parameters.

Here, we can calculate that

$$p = \frac{1}{1 + b^{-(\beta_0 + \beta_1 x_1 + \beta_2 x_2)}},$$

is the probability of Y = 1.

The formula shows that once $\beta_i$ are fixed, we can effortlessly compute the probability that gives Y = 1 for a given observation. The main purpose of a logistic model is to calculate observation $(x_1, x_2)$ and estimate the probability that $P(Y = 1)$. Logistic regression, which is a general statistical model, was originally developed and introduced by Joseph Bergson [22].

### 4.2.2. Naive Bayes Classifier

Naive Bayes classifiers are based on "naive" independence assumptions between the features used to construct classifiers, i.e., models that assign class labels to problem instances, where the class labels are drawn from some finite set [23]. Maximum-likelihood training can be performed by evaluating the number of occurrences of the sample [24], which requires linear time, rather than expensive iterative approximation (as used for many other types of classifiers).

In our context, the goal of naive Bayes classification is to find the probability that data $x$ belongs to indoor class I or outdoor class O based on Bayes' theorem.

$$P(A|B) = \frac{P(B|A)P(B)}{P(A)}. \tag{2}$$

The probability of $x$ belonging to the indoor or outdoor class is given as follows.

$$P(y = I|x); \ P(y = O|x). \tag{3}$$

We can find the class of data $x$ by selecting the class with the highest probability.

$$c = \underset{c \in \{I,O\}}{arcmax} p(c|x). \tag{4}$$

By applying naive Bayes to Equation (4), we can obtain $c$ as follows.

$$c = \underset{c \in \{I,O\}}{arcmax} \frac{p(x|c)p(c)}{p(x)}, \tag{5}$$

$$c = \underset{c \in \{I,O\}}{arcmax} p(x|c)p(c). \tag{6}$$

In addition, by applying the assumption that all features in $x$ are independent, we can calculate $c$ as follows.

$$c = \underset{c \in \{I,O\}}{arcmax} p(x_1, x_2 \dots, x_9|c)p(c) c = \underset{c \in \{I,O\}}{arcmax} \prod_{i=1}^{9} p(x_i|c) \, p(c). \tag{7}$$

In the test phase, the class of new data $x$ is found as follows.

$$c = \underset{c \in \{I,O\}}{arcmax} \prod_{i=1}^{9} p(x_i|c) \, p(c). \tag{8}$$

Despite their naive design and apparently oversimplified assumptions, naive Bayes classifiers have worked quite well in many complex real-world situations. Due to the "naive" independent assumption between the features, the naive Bayes classifier is simple and provides fast coverage during the training phase. It can also solve pattern recognition problems with a small amount of training data

to estimate the parameters required for classification. However, naive Bayes cannot work with data that never happened before, and the "naive" assumption is often impractical for many real-world problems.

### 4.2.3. Support Vector Machine

SVMs are supervised learning models that analyze data used for classification and regression analysis. An SVM training technique assigns all new examples to a single category, thereby making it a non-probabilistic binary linear classifier. An SVM model is a representation of the examples in space that maps the examples of separate categories divided by a clear gap calculated to be as wide as possible. All new examples are mapped into that same space and predicted to belong to a category based on which side of the gap they fall. In addition to linear classification, SVMs can classify nonlinear data efficiently using a kernel trick, i.e., mapping their inputs to high-dimensional feature spaces.

In the SVM technique, the training set for the I/O detection problem is $(x_1, y_1), (x_2, y_2), \ldots, (x_{850}, y_{850})$, where $x_i \in R^9$ are the input data and $y_i$ is the class of $x_i$. The class of the input data is selected based on the sign of $y_i$ ($y_i = 1$ when $x_i$ is outdoor, and $y_i = -1$ when $x_i$ is indoor). In the training phase, the I/O detection program finds hyperplane that classifies the two classes. The hyperplane is expressed as follows.

$$\mathbf{w}^{\mathrm{T}}\mathbf{x} + b \;=\; w_1 x_1 + w_2 x_2 + \cdots + w_9 x_9 + b \;=\; 0. \tag{9}$$

With a specific data point $(x_i, y_i)$, the distance to the hyperplane is calculated as follows.

$$\frac{y_n(\mathbf{w}^{\mathrm{T}}\mathbf{x}_n + b)}{\|w\|_2}. \tag{10}$$

Since Equation (10) is positive, we can find the margin of a data point to hyperplane as follows.

$$\mathrm{margin} \;=\; \min_n \frac{y_n(\mathbf{w}^{\mathrm{T}}\mathbf{x}_n + b)}{\|w\|_2}. \tag{11}$$

The training phase is finding the margin as large as possible.

$$(\mathbf{w}, b) \;=\; \operatorname*{argmax}_{w,b}\left\{\min_n \frac{y_n(\mathbf{w}^{\mathrm{T}}\mathbf{x}_n + b)}{\|w\|_2}\right\} \;=\; \operatorname*{argmax}_{w,b}\left\{\frac{1}{\|w\|_2}\min_n y_n(\mathbf{w}^{\mathrm{T}}\mathbf{x}_n + b)\right\}. \tag{12}$$

Equation (12) can be simplified as follows.

$$(\mathbf{w}, b) \;=\; \operatorname*{argmin}_{w,b} \frac{1}{2}\|w\|_2^2 \ \text{subject to}:\ 1 - y_n(\mathbf{w}^{\mathrm{T}}\mathbf{x}_n + b) \leq 0,\ \forall n \;=\; 1, 2, \ldots, N. \tag{13}$$

We find the weigh matrix of Equation (13) as follows.

$$\mathbf{w} \;=\; \sum_{n\,=\,1}^{850} \lambda_n y_n \mathbf{x}_n \ \text{where } S \;=\; \{n : \lambda_n \neq 0\}, \tag{14}$$

$$b \;=\; \frac{1}{N_S}\sum_{n\in S}(y_n - \mathbf{w}^{\mathrm{T}}\mathbf{x}_n) \;=\; \frac{1}{N_S}\sum_{n\in S}\left(y_n - \sum_{m\in S}\lambda_m y_m \mathbf{x}_m^T \mathbf{x}_m\right). \tag{15}$$

SVMs are one of the most popular classifier techniques and are highly effective when dealing with unstructured and multidimensional input data. SVMs are very flexible, and we can use an SVM even if we do not possess knowledge of the data structure. However, SVMs are very complicated and provide slow coverage. SVM models also are hard for hyperparameters tuning.

### 4.2.4. K-Nearest Neighbors Classifier

The KNNs technique is a nonparametric method used for both classification and regression [25], where the input consists of training examples in the feature space. The KNN technique is one of the simplest of all machine learning techniques, in which it only approximated locally, and all computations are calculated in the classification state. Here, an object is classified by the majority of its neighbors, where the object is assigned to the most common class among its nearest neighbors.

Due to the simplicity of the technique, KNN is very easy to implement; however, it can still handle unstructured data and simple valuation with good estimation of the test set. With lazy learning, the KNN calculation is performed in the test phase; thus, with a large dataset, the test phase requires more time than other machine learning techniques. This characteristic makes offline learning impossible for KNN. Another drawback of KNN is its sensitivity to noise, which makes it unattractive for complex problems.

### 4.2.5. Neural Network

Artificial neural networks (ANN) are computing systems that are including many different machine learning techniques to work together and process complex data inputs. Such systems "learn" to perform tasks using examples, and they increase performance gradually without being programmed according to any task-specific rules. Instead, they automatically generate identifying characteristics from the learning material they process. The original goal of the ANN approach was to solve problems in the same manner a human brain would. However, ANNs have been developed over time to perform specific tasks, leading to deviations from biology. ANNs have been used in a variety of fields, including computer science, security, medicine, social network science, board and video games, and military purposes.

In this study, the neural network model was designed as follows: nine for input layer (each receiving a GPS signal), 18 hidden neural, and two output neural (Figure 10).

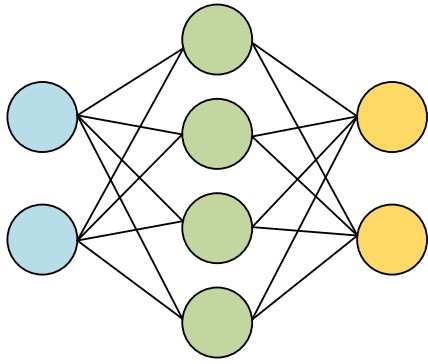

Input layer    Hidden layer    Output layer

**Figure 10.** Neural network model.

### 4.3. Training Phase

Since the input data are well collected, the training phase is easier, which leads to high accuracy in the model. Errors will fall to the semi-indoor environment, where the GPS signal is still strong despite the indoors environment. This is the characteristic of the input data, yet the learning model can reduce the impacts to minimum.

We employed L2 norm for error calculation, where errors are calculated as the Euclidean distances between two points. Note that L2 norm is the best choice for multidimensional input vector data.

Based on the required accuracy and user devices, we can also apply offline training. Offline training is suitable for low-accuracy applications or devices that do not require permanent mobility

because it does not require significant resources, has a fast calculation process, and does not require an update operation. Offline learning is also an advantage of this scheme because the final model is very simple, offers fast calculation, and requires only the current GPS signal information. In contrast, online training can increase the accuracy of the model; however, online training results in a complex model and requires extra calculations to update the model. Online training is more suitable for mobile applications, where the device's resources are available and applications requires high accuracy.

In this study, we focus on offline training, and we tested with many machine learning techniques to identify the best solution for the GPS signal classifier.

### 4.4. Testing Phase

Generally, the preprocessing phase improves the quality of the input data; thus, the machine learning models can work effectively with very high accuracy. First, we had a training set with 423 indoor samples, 427 outdoor samples, and a small number of semi-indoor samples that we set as indoor samples. Note that we selected this training set to simulate normal activities (most time is spent in indoor and outdoor environments rather than semi-indoor environments). With the test set with only indoor and outdoor samples, all machine learning models reach very high accuracy (minimum was 96% with the SVM technique, and the maximum was 98% with the KNN technique). To facilitate better evaluation of the data set, we increased the number semi-indoor environment samples as an individual test set to compare the accuracy of all models in a high noise environment. This test set included 170 samples (70 indoor samples, 50 outdoor samples, and 50 semi-indoor samples categorized as indoor samples). We also proposed a unique preprocessing method to improve the final result of this test set.

We employed an offline learning method for I/O detection. Offline learning uses the trained model for the actual application, and it updates the model based on the programmer's schedule. Offline learning provides a simple model with fast calculations. The application does require a large amount of training data, as well as learning phase calculation. Another advantage of offline training is the uniformity of all applications: note that the model will do the same classification for all GPS sensors.

## 5. Evaluation

### 5.1. Testing with Indoor/Outdoor Sample Test Set

Figure 11 shows the accuracy of the test set with indoor and outdoor samples. We also included I/O detection using a combined sensor [1,11] for comparison. The first test set included 70 indoor and 50 outdoor samples, with high differences in both the number and strength of the received GPS signal. The proposed scheme demonstrated an impressive result, with the highest overall rate of 98% and lowest of 96%. The results indicate that the strength of the GPS signal from the smartphone sensor can be used to extract features to be used in the I/O detection classifier, and the method can be used for other applications. Note that the sample data can be representative for the overall feature of indoor/outdoor areas because the GPS signal is essentially the same at other locations on Earth. With the indoor/outdoor test set, the I/O detection application can obtain impressive accuracy, even without performing a preprocessing phase. This high accuracy is due to the large bias of the indoor and outdoor environments, in which the GPS signal is very weak inside buildings. However, this test set is not included the semi-indoor environment, which is the most difficult environment for I/O detection; therefore, the application does not promise smooth transfer from an outdoor environment to an indoor environment (and vice versa). Thus, another test set for semi-indoor environments was added and evaluated with the indoor/outdoor environment test set. Here, we tested using the semi-indoor environment and set the samples to indoor samples.

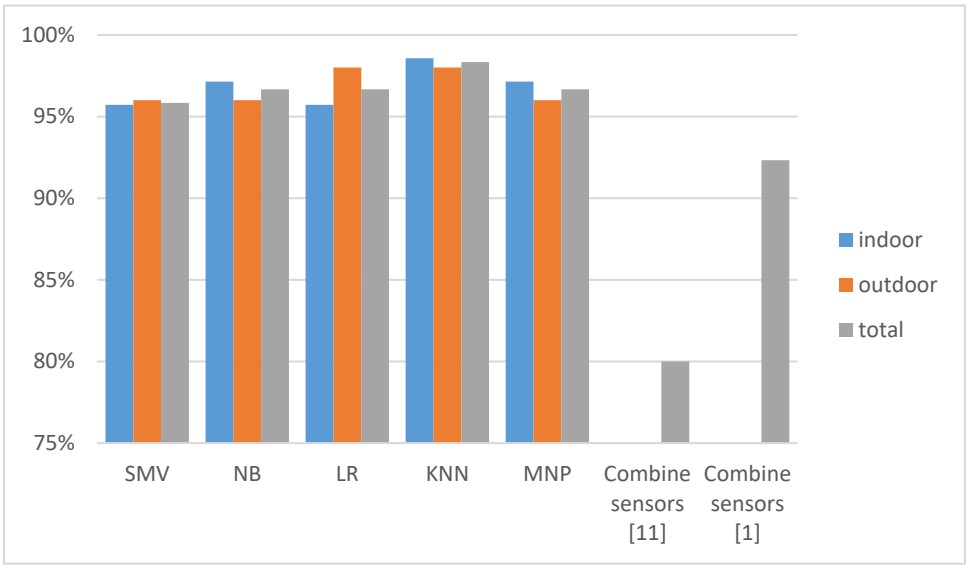

**Figure 11.** Indoor/outdoor (I/O) detection accuracy.

## 5.2. Testing with Indoor/Outdoor and Semi-Indoor Sample Test Set

This test set was included to improve the application's ability to identify an indoor environment even if the GPS signal was weaker due to the semi-indoor environment. This characteristic is extremely helpful when moving from an outdoor environment to an indoor environment (and vice versa). The second test set included 70 indoor, 50 outdoor, and 50 semi-indoor samples. For the comparison, we also included I/O detection using a combined sensor [9] for comparison. Figure 12 shows the accuracy for indoor and outdoor cases separately, in which the semi-indoor environment is categorized as indoor. The accuracy of indoor detection was reduced significantly, which indicates confusion between semi-indoor and outdoor environments in the classifier phase. The result of indoor environment detection was further analyzed to identify the root causes of the poor performance of several classifiers. In consideration of the lower accuracy compared to the outdoor environment, the indoor and semi-indoor samples were misclassified, which indicates that the GPS signal strength of semi-indoor cases is similar to outdoor cases, and it is very difficult to distinguish these cases. In many semi-indoor environments, there is very little change in the GPS signal compared to outdoors, which leads to a malfunction in the I/O detection application. The noise added to the GPS signal in an outdoor environment is also a reason for errors because this makes the GPS signal become weaker and confused with the semi-indoor signal, which decreases the accuracy of the learning model.

Compared to the combined sensor technique, the accuracy of the proposed scheme is still better (95% for the KNN technique compared to 88% for the combined sensor [9]). This emphasizes the effectiveness of the GPS-based I/O detection scheme. However, the result is insufficient, and we can improve it by employing a unique preprocessing process.

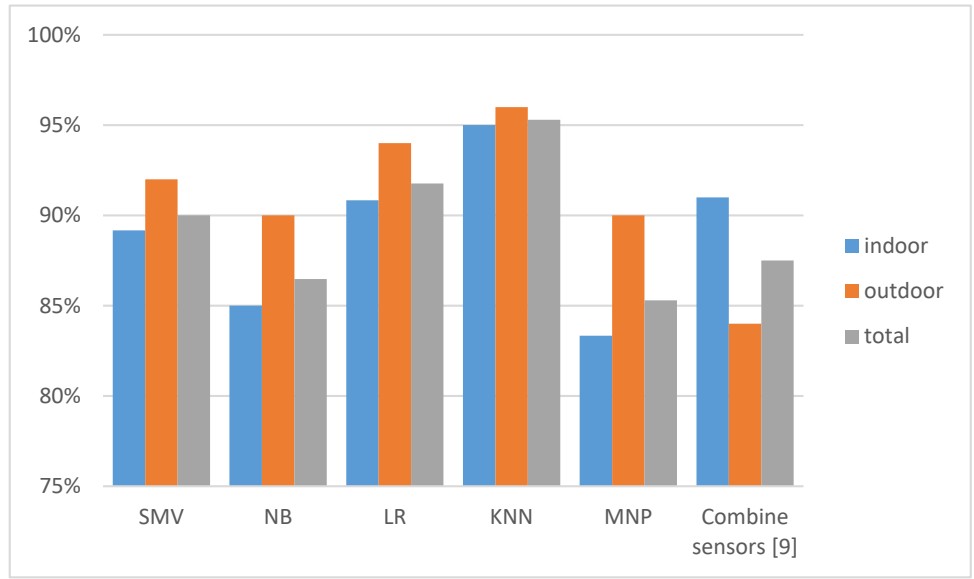

**Figure 12.** Accuracy of I/O detection with semi-indoor samples (classified as indoor samples).

### 5.3. Proposed Preprocessing Process

For the dataset including 1020 individual data, each of them was based on the strength of GPS signals corresponding to a specific place. All possible GPS signals were collected, and, using the mobile application, the analog input was transformed to digital data for further analysis. Here, the data included information about the GPS signal (zero to 10 features). We denoted the $i$th data as $X_i$, and the GPS signal from satellite $j$ in $X_i$ data as $x_{ij}$. We regularized the input data as follows.

Step 1 Reduce the number of GPS signals to nine.

The tenth GPS appears very infrequently, and, as the GPS signal is very small, this information is not useful for analysis.

Step 2 Fill the missing datum.

As mentioned previously, low-strength GPS signals are removed by the mobile application. Thus, if the GPS signal is too weak (in a building, to be blocked by obstacle), nine GPS signals cannot be obtained. The datum will be filled by zeroes; thus, these data will not affect the strongest signals, which contain the most information about I/O detection.

Step 3 Rearrange data for further analysis.

The strong signals contain the most information about I/O detection; however, the order of these signals is chaotic and independent from each other. Thus, we cannot find any pattern because the strong signals mix with the weak signal, which makes the entire dataset chaotic and unpredictable. Therefore, we propose a simple solution for this problem, i.e., we rearrange the GPS signals from strongest to smallest. As a result, strong GPS signals will take priority during the training and testing phases. Even if this does not work effectively, we can add a heavyweight to strong GPS signals before applying provided schemes.

The preprocessing sequence is shown in Figure 13.

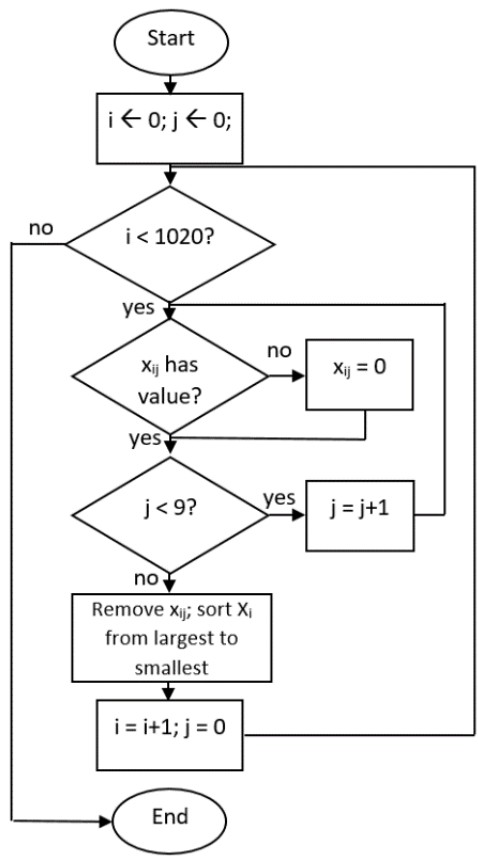

**Figure 13.** Preprocessing flow.

*5.4. Testing with Indoor/Outdoor and Semi-Indoor Sample Test Set with Proposed Data Processing*

Figure 14 shows the accuracy for indoor and outdoor cases obtained after applying the proposed preprocessing. The preprocessing phase increases the I/O detection accuracy significantly (approximately 4% for the worst case and 2% for the best case). The machine learning techniques clean the input data, as well as propose a better structure for the learning model.

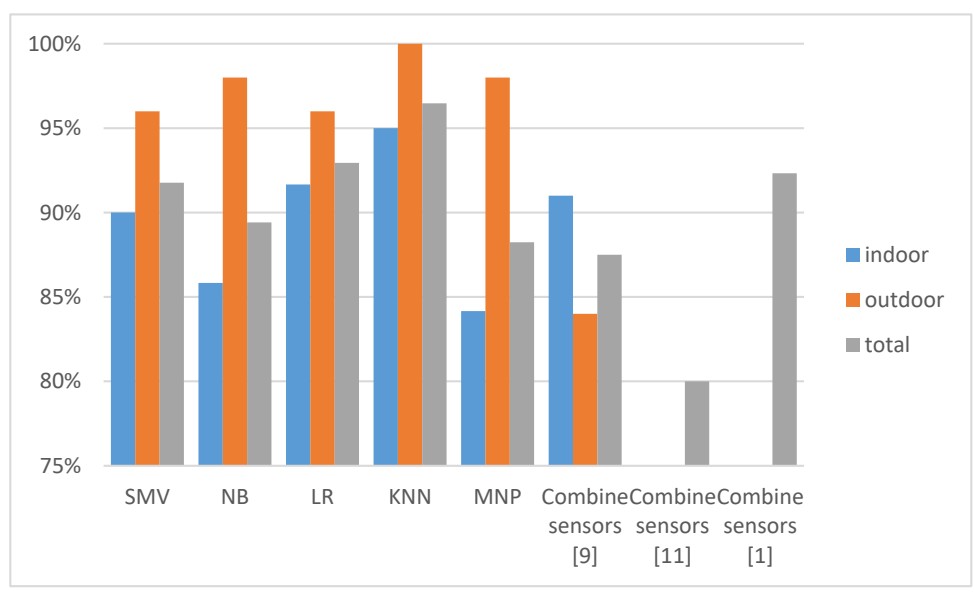

**Figure 14.** I/O detection accuracy obtained with proposed preprocessing.

The performance of the proposed approach was compared to traditional technologies commonly used for I/O detection. The first comparison was against I/O detection using GPS. The proposed approach was compared to three different studies [1,9,11]. The proposed method is clearly better because it uses more information, has better preprocessing, and applies suitable Machine Learning model.

Based on the overall scheme, other classifier methods, such as LSTM and Meta-classifier are applied. However, they are not suitable for the I/O detection scheme and the accuracy is also lower than KNN. In proposed scheme, I/O detection focused on the simple and effective methods, as it does not require mobile resources. Based on this approach, the machine learning methods are more suitable for the scheme. In contrast, both LSTM and Meta-classifier required large resources for both training and testing phases, so that they are not considered as promised methods for proposed scheme. Another reason is that the accuracy of LSTM and Meta-classifier is lower than KNN. LSTM reaches 91% overall accuracy and Meta-classifier is 94%, which cannot be considered in the proposed scheme.

The machine learning model will be evaluated based on the accuracy of test sets, as well as training coverage. We use offline training for the application; thus, training coverage may not be important. Nonetheless, it will remain a criterion used to evaluate the machine learning model.

## 6. Conclusions

In this paper, we proposed an indoor and outdoor environment detection scheme that classifies indoor and outdoor environments at high accuracy under a simple implementation. We also proposed a data preprocessing technique. Using the proposed data preprocessing process and suitable machine learning techniques, the proposed I/O detection scheme obtains quick and accurate detection results in various time and environments. We comprehensively tested I/O detection through a prototype implementation and evaluated the system based on different data samples collected in the Kookmin University area. The prototype systems obtained an accuracy of 97%. We particularly conducted a case study where we made use of I/O detection results to infer the GPS availability and accuracy under various indoor/outdoor environments.

**Author Contributions:** All authors contributed to this paper: Van Bui and Nam Tuan Le proposed the idea and implementation methodology, reviewed and edited paper; Van Bui performed all experiments and wrote the paper, verified the experiment process and results; Thanh Luan Vu collected data and created the first version of data augmentation software; Van Hoa Nguyen proposed and optimized machine learning techniques, performed parts of experiments; and Yeong Min Jang supervised the work and provided funding support. All authors have read and agreed to the published version of the manuscript.

**Funding:** This research was financially supported by the Ministry of Trade, Industry and Energy (MOTIE) and Korea Institute for Advancement of Technology (KIAT) through the International Cooperative R&D program.

**Conflicts of Interest:** The authors declare no conflict of interest.

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
