# Peer review of "GPS-Based Indoor/Outdoor Detection Scheme Using Machine Learning Techniques"

_applsci, doi:10.3390/app10020500_

Round 1

Reviewer 1 Report

The research paper studies an indoor-outdoor detection scheme based on machine learning reasoning. Inference models are fed with GPS traces either indoor, or outdoor or semi-indoor. Comparison is based on certain machine learning models which reach an acceptable prediction accuracy value. However, some issues are not clear. Specifically:

For outdoor and marginally semi-indoor situations GPS traces are reliable. In general, this is not the case for indoor values where obstacles are present. How the model treats the cases of indoor positioning in case of low signal?  Why not to use a more advanced neural network model like LSTM neural network which is more efficient than simple neural networks? Why not to use a meta classifier, like voting schema, to evaluate your model since meta classifiers in general increase the prediction accuracy? I would expect a McNeamar's test to prove the statistical significance of the selected classifier.

Author Response

We are very thankful to the reviewer for pointing out few things to improve the content of paper. We have tried our best to improve the paper according to the reviewer’s suggestions.

All changes and responses made according to reviewer’s comments are listed in the attached file.

Reviewer 2 Report

The submitted manuscript shown a proposal for detecting indoor and outdoor environment. The analysis of the proposal is based on experimental GPS and sensor data collected in different working condition. The obtained results show a significant success rate that could be higher than 90%.

The main issue of the proposal is that it does not consider the most challenging environment in which the indoor/outdoor discrimination is essential. For example, in emergency scenarios could include satellite obstructed environments in which the reduced GPS signal strength could be misinterpreted. The author are encouraged to refer to the paper DOI: 10.1109/JSYST.2015.2462742 in order to get some insights for improving the manuscript, at least in the Related Work section. In particular it was shown that the joint processing of WiFi and GPS signals can produce significant benefits that are worth of consideration.

Another aspect that could be considered is the joint usage of GPS and other satellites of different navigation systems such as Galileo.  For more information the paper DOI: 10.1109/RAST.2005.1512598 is suggested.

Another aspect to be made clear is the needed volume of training data sets in order to make the proposal working. Which application types are compliant with this requirement? Could any stationarity issues be found?

Finally, the authors could also investigate if the proposal can be applied to vehicular application, when vehicles access urban canyons that could significantly impair signal reception from navigation satellites.

Author Response

We are very thankful to the reviewer for pointing out few things to improve the content of paper. We have tried our best to improve the paper according to the reviewer’s suggestions.

All changes and response made according to reviewer’s comments are listed in the attached file.

Round 2

Reviewer 2 Report

The effort done by the Authors for addressing the reviewer's suggestions on a point-by-point basis is appreciated, as also highlighted in the specific authors's reply to reviewers. The revised paper is therefore a sufficiently improved version of the originally submitted  paper. Only a stylistic comment in the writing of the references persists. All the authors's last name should be written in full. Only the first name can be abbreviated by writing the first letter with a dot.

Author Response

We are very thankful to the reviewer for pointing out the references errors  to improve the content of paper. We have tried our best to improve the paper according to the reviewer’s suggestions.

All changes and response made according to reviewer’s comments are marked in the updated paper.
